# A Bio-Inspired Spiking Neural Network with Few-Shot Class-Incremental Learning for Gas Recognition

**DOI:** 10.3390/s23052433

**Published:** 2023-02-22

**Authors:** Dexuan Huo, Jilin Zhang, Xinyu Dai, Pingping Zhang, Shumin Zhang, Xiao Yang, Jiachuang Wang, Mengwei Liu, Xuhui Sun, Hong Chen

**Affiliations:** 1School of Integrated Circuits, Tsinghua University, Beijing 100084, China; 2Suzhou Huiwen Nanotechnology Co., Ltd., Suzhou 215004, China; 3State Key Laboratory of Transducer Technology, Shanghai Institute of Microsystem and Information Technology, Chinese Academy of Sciences, Shanghai 200050, China

**Keywords:** spiking neural network, few-shot learning, incremental learning, gas recognition

## Abstract

The sensitivity and selectivity profiles of gas sensors are always changed by sensor drifting, sensor aging, and the surroundings (e.g., temperature and humidity changes), which lead to a serious decline in gas recognition accuracy or even invalidation. To address this issue, the practical solution is to retrain the network to maintain performance, leveraging its rapid, incremental online learning capacity. In this paper, we develop a bio-inspired spiking neural network (SNN) to recognize nine types of flammable and toxic gases, which supports few-shot class-incremental learning, and can be retrained quickly with a new gas at a low accuracy cost. Compared with gas recognition approaches such as support vector machine (SVM), k-nearest neighbor (KNN), principal component analysis (PCA) +SVM, PCA+KNN, and artificial neural network (ANN), our network achieves the highest accuracy of 98.75% in five-fold cross-validation for identifying nine types of gases, each with five different concentrations. In particular, the proposed network has a 5.09% higher accuracy than that of other gas recognition algorithms, which validates its robustness and effectiveness for real-life fire scenarios.

## 1. Introduction

With the development of artificial intelligence and sensing technology, the electronic nose (E-nose) is becoming increasingly intelligent. As a bionic detection technique that mimics the mammal’s olfaction system well [1], the E-nose has been developed for nearly 40 years [2] and is remarkably successful in many applications including food safety [3,4,5], environment monitoring [6], agriculture [7], industry [8], public safety [9,10] and so on.

Generally, the E-nose is composed of two parts (shown in Figure 1): a gas sensing system and an information processing system [11,12]. The “gas sensing system” obtains gas data from the gas sensor array, and the “interface readout circuit” converts the sensor signals into electrical signals. Finally, the “gas recognition circuit” outputs the final recognition results with gas recognition algorithms. In the E-nose system, gas recognition algorithms play an important role as they determine the decision-making result of the system. As reported in [13], for E-nose systems it is vital to improve the gas qualitative recognition accuracy; if the gas type is recognized incorrectly, gas concentration quantitative estimation is meaningless. As said in [14], the reasonable improvement of the gas recognition algorithms is an important support for the development of the E-nose system.

Many gas recognition algorithms have been proposed, including k-nearest neighbor (KNN) [15,16], support vector machine (SVM) [17,18], decision tree (DT) [19], and naive Bayesian model (NBM) [20]. However, these algorithms are able to identify less than five kinds of gases as reported, which is not applicable for real-time multi-gas recognition tasks. As we know, in fire various toxic gases will be generated before and after the explosion of flammable and explosive gases, which requires identifying a wide range of gases, including flammable, explosive, and toxic gases. Unfortunately, there are no distinct examples to offer evidence that these common gas recognition algorithms are able to identify multiple gases [21].

On the other hand, with the rapid development of the artificial neural network (ANN), it is proved that an increasing number of ANNs including the back propagation neural network (BPNN) [22], convolutional neural network (CNN) [23], and radial basis function neural network (RBFNN) [24] perform well in gas recognition. In 2021, D. Ma et al. [25] proposed a 15-layer CNN to recognize 10 different VOCs (volatile organic compounds) with 92% accuracy. Additionally, the spiking neural network (SNN) has also been developed for gas recognition in recent years. In 2020, Imam Nabil and Cleland T.A [26] presented an SNN with one-shot learning and identification of odorant samples. The two-layer network with 72 channels reached 92% accuracy for 10 kinds of gas recognitions under 40% noise. However, this work needs high-dimensional gas sensor data, which are difficult to implement in the wild.

One fundamentally ill-posed problem that needs to urgently be addressed is sensor drift, with which the sensitivity and selectivity profiles of gas sensors gradually change over the time of use or disuse [27]. Fortunately, SNNs have a good anti-noise ability [28], and are robust to the gas data easily affected by surroundings like temperature and humidity. In addition, SNN-based biological plausible learning methods, like STDP, perform well in online learning [26,27], which is meaningful for gas recognition. Moreover, in [27] the SNN exhibits 94.93% classification accuracy after few-shot learning without catastrophic forgetting, which is a challenge for most ANNs [29].

In this paper, we propose a novel bio-inspired SNN with few-shot class-incremental learning for gas recognition from 24 sensors. The network is able to recognize over nine types of gases commonly found in fires with five different concentrations. Meanwhile, it achieves at least 98.75% classification accuracy with only five training samples per gas, and retrains the network with new gases with little accuracy loss. Additionally, due to the characteristics of SNNs, a lot of multiplication is avoided. Sparse synapse connections further reduce the calculation cost.

The rest of the paper is organized as follows: In Section 2, we introduce the gas data acquisition experiments. The proposed SNN is presented in Section 3, and the experimental results are discussed in detail in Section 4. Section 5 concludes the paper.

## 2. Experiments

We collected gas data through an E-nose gas sampling system from Suzhou Huiwen Nanotechnology Co., Ltd. (Suzhou, China) The system consists of 24 gas sensors, each of which is sensitive to different gases. With the fire station’s recommendation, we chose nine types of flammable and toxic gases common in fires: formaldehyde, ethanol, propane, methanol, methane, carbon monoxide, acetone, hydrogen sulfide, and ammonia, as illustrated in Table 1. Each type of gas was tested 25 times for five different concentrations, thereby 225 samples were obtained of nine categories.

The gas sample collecting process of the E-nose system has four steps: (1) preparation step: the E-nose system works at a supply voltage of 3.3 V and the sampling frequency is set to 1 Hz. In the test environment, the temperature is set in the range of 25.6~27.9 °C and the relative humidity is between 40~50%. (2) baseline test step: it takes 15~30 min for the system to fill with pure air until all 24 gas sensors reach their steady value $V_{steady0},\ldots ,V_{steady24}$. (3) target gas test step: Continuously injecting the target gas into the E-nose sample system and waiting for about 5 min until all 24 sensors reach their steady response value $V_{response0},\ldots ,V_{response24}$. (4) baseline recovery step: injecting clean air to extract the target gas from the system and waiting for about 10~15 min until all 24 gas sensors return to the original baseline value $V_{steady0},\ldots ,V_{steady24}$.

If we take 2000 ppm methane (CH4) as an example, its response curves of the 24 gas sensors are represented in Figure 2, from which we can see that in the baseline test step and the response curves $V_{steady0},\ldots ,V_{steady24}\left(t\right)$ are stable. When we continuously inject the target gas (2000 ppm methane), the response voltages ($V_{response}\left(t\right)$) of the 24 sensors are different, as shown in the middle of the response curve in Figure 2. Finally, when we inject the clean air, the response voltages return to the baseline value. According to the study [30], we use the differential method to process the sensor data and reduce the fixed external interference δ caused by noise and drift as described in Formula (1), where i=1,2,…,24.
(1)$$\;V_{outputx}\left(t\right)=\left(V_{responsex}\left(t\right)+\delta \right)-\left(V_{steadyx}\left(t\right)+\delta \right)=V_{responsex}\left(t\right)-V_{steadyx}\left(t\right)$$

## 3. Methods

### 3.1. Signal Preprocessing

As the gas data collected by the E-nose system are inevitably interfered by the environment and sensor drift, to ensure reliable classification performance, the gas data are pre-processed to alleviate the interference. We propose a novel preprocessing algorithm to normalize the gas sensor data to match the proposed SNN.

First, we adopt the variant of Minimum-Maximum Normalization (MMN) for gas data scaling to compensate for the heterogeneity of the 24 different gas sensors. The maximum and minimum response of each sensor can be easily found from the gas dataset collected by the E-nose system. Let $G_{1},\;G_{2},\;\ldots ,\;G_{n}$ be the responses of n sensors to a given gas, and $M_{1},\;M_{2},\;\ldots ,\;M_{n}$ and $S_{1},\;S_{2},\;\ldots ,\;S_{n}$ be the maximum and minimum response values of the n sensors, respectively. Next, the set *S*
$:\left\{\rho \frac{G_{1}-S_{1}}{M_{1}-S_{1}},\rho \frac{G_{2}-S_{2}}{M_{2}-S_{2}},\;\ldots ,\;\rho \frac{G_{n}-S_{n}}{M_{n}-S_{n}}\right\}$ are deemed as the sensor-scaled values, and we keep γ maximum elements among the set *S* where ρ and γ are the scaling factor and cutoff factor, respectively. With this we aim to eliminate the interference caused by the insensitivity of the sensors to a given gas. It is proven that this step could improve the recognition performance significantly (Section 3.4). 

### 3.2. Neuron Model

In order to model neuron dynamics, we adopt the integrate-and-fire (IF) model to simulate the basic functions of biological neurons, which are described in Equation (2):(2)$$\tau \frac{dV\left(t\right)}{dt}=(V_{rest}-V\left(t\right))+\tau_{exc}V_{in}\left(t\right)-\tau_{inh}V_{in}\left(t\right)\;$$
where $V\left(t\right)$ is the membrane voltage, $V_{rest}$ is the resting membrane voltage, $V_{in}\left(t\right)$ is the changing voltage caused by input spikes, $\tau_{exc}$ and $\tau_{inh}$ are conductance constants of excitatory and inhibitory synapses, respectively, and τ is the timing constant. When the spike $S_{exc}$ from excitatory synapses arrives at the neuron model, $\tau_{exc}V_{in}\left(t\right)$ causes an increase in the membrane voltage $V\left(t\right)$. On the contrary, when $S_{inh}$ from inhibitory synapses arrives, $V\left(t\right)$ decreases. Once $V\left(t\right)$ reaches a membrane threshold $V_{th}$, the neuron model fires and $V\left(t\right)\;$ is set to $V_{rest}$ immediately. Then, the neuron model is in a refractory period, in which it cannot receive any spikes from any neurons or itself spike.

### 3.3. Network Architecture

The proposed network consists of the two layers shown in Figure 3. The first layer is the excitatory layer containing 24 neurons, where each is connected to one gas sensor. The second layer is the inhibitory layer including 120 neurons, where every 5 neurons correspond to one excitatory neuron. Each excitatory neuron and its corresponding five inhibitory neurons are regarded as a group called a glomerulus, hence there are 24 glomeruli in this model. Each glomerulus aims to increase the spiking levels of its own excitatory neuron inside and decrease the spiking levels of excitatory neurons from the other glomeruli.

We adopt a spike precedence coding method [31] for the excitatory neurons in the first layer, which is timing-based. With the method, the stronger input of sensor, the earlier spike phase. As described in [26], this spike timing-based coding offers considerable speed and efficiency advantages. After signal preprocessing, the gas data are transformed into spikes with spike precedence coding, and the number of spikes is proportional to the $V_{outputx}\left(t\right)$ of the gas sensor.

The connection between the excitatory and inhibitory neurons in the same glomerulus is called excitatory synapses, the connection probability is chosen to be 0.80 based on experiments in our work. The inhibitory neurons are randomly connected to the other excitatory neurons in different glomeruli with a connection probability of 0.25, in our case, and the connections are called inhibitory synapses. The excitatory and inhibitory synapses are adjusted by the conductance constants $\tau_{exc}$ and $\tau_{inh}$, respectively.

The inhibitory synaptic connection causes lateral inhibition and competition among the excitatory neurons. To ensure the lateral inhibition is neither too weak, which means it does not have any influence on the weight update in the training stage, nor too strong, which leads to an imbalance between the winner and loser neurons, we set a ratio $\varphi =\frac{\tau_{exc}}{\tau_{inh}}$ between the excitatory and inhibitory synapse to adjust the connection strength. 

### 3.4. Learning Rule

Spike-Timing-Dependent Plasticity (STDP) is adopted as the learning rule for both excitatory synapses and inhibitory synapses. STDP is a local learning method, where the synaptic plasticity is sensitive to the spiking time of the pre- and post- neurons. We have adopted the exponential variant STDP in (3) to update the weights of synapses through the spike-timing difference between the postsynaptic inhibitory and the presynaptic excitatory neurons. When the spike time t1 of the post-neuron is earlier than the spike time t2 of the pre-neuron (t1<t2), the weight w will increase by ∆w. Otherwise, the weight w will decrease by ∆w. The variable ∆w is described below:(3)$$\;{∆}_{w}=\left\{\begin{array}{c} \eta_{exc}e^{\frac{t_{2}-t_{1}}{\varepsilon_{exc}}}\;\left(t1<t2\right)\\ -\eta_{inh}e^{\frac{t_{2}-t_{1}}{\varepsilon_{inh}}}\;\left(t2<t1\right) \end{array}\right.\;$$
where $\eta_{exc}$ and $\eta_{inh}$ are learning rates of excitatory and inhibitory synapses, respectively, and $\varepsilon_{exc}$ and $\varepsilon_{inh}$ are used to measure the balance between the excitation degree caused by excitatory synapses and the inhibition degree caused by inhibitory synapses, respectively.

Here, we apply the STDP rules with lateral inhibition in the training stage between excitatory and inhibitory layers. The weight of the excitatory neuron $EN_{i}$ in the excitatory layer and the inhibitory neuron $IN_{j}$ in the inhibitory layer are as follows:(4)$$\;W_{ENi}\left(t\right)=W_{ENi}\left(t\right)+\underset{j\in P_{s}\left(EN_{i}\right)}{\sum }∆w_{EN_{i}}^{\left(j\right)}\left(t\right)*spike{\left(t\right)}^{\left(j\right)}\;$$
(5)$$\;W_{INj}\left(t\right)=W_{INj}\left(t\right)+∆w_{INj}{}^{\left(j\right)}\left(t\right)*spike{\left(t\right)}^{\left(i\right)}$$ where the set $P_{s}\left(EN_{i}\right)$ includes all inhibitory neurons INs that have inhibitory synaptic connections to the excitatory neuron $EN_{i}$. The $spike{\left(t\right)}^{\left(j\right)}$ is a spike function, whose value is 1 when the neuron $IN_{j}$ spikes with timestamp t, otherwise its value is 0.

If the input stimulus is strong enough to enable the neuron $EN_{i}$ to spike, the membrane voltage will rise for all $IN_{j}$ neurons that have excitatory synaptic connections in the same glomerulus (as shown in Figure 3). When the neuron $IN_{j}$ spikes (as described in Formula (5)), the weight of other excitatory neurons in different glomeruli in the form of inhibitory synapses will be updated according to Formulas (3) and (4). After several epochs, we find the excitatory neurons with a strong input stimulus become more active, and others relatively inactive, which is similar to the winners-take-all mechanism. As a result, the relatively active excitatory neurons are prominent so as to obtain good classification results.

### 3.5. Training and Output Classifier

To train the proposed network, we collected 9 types of gases. Each type of gas has 5 different concentrations, and each gas was tested five times with each concentration. Therefore, we obtained 225 gas samples. Generally, the ratio between the training and test sets is 0.8. As the proposed algorithm supports few-shot learning, we set the ratio to be 0.2. Each sample for the training set is calculated by 100 iterations. For comparison, the spiking times of the ENs in the excitatory layer are inputs to the classifier for sample classification according to a Manhattan distance metric. We measure the classification performance by computing the similarity between the output of the training sample ($V_{train}$) and test sample ($V_{test}$). The test samples are classified according to the identity of the training samples that are most similar to them.

We adopt a Pearson correlation coefficient (PCC) to assess the degree of correlation between the training and test output vectors, $V_{train}$ and $V_{test}$. For n gas sensors, $V_{train}$ and $V_{test}$ are n-dimensional vectors. We define the PCC as follows (6)–(11):(6)$$PCC=\frac{P_{xy}}{P_{x}*P_{y}}\;$$
(7)$$P_{xy}=\frac{1}{n-1}\overset{n}{\underset{i=1}{\sum }}\left(x_{i}-\bar{x}\right)\left(y_{i}-\bar{y}\right)\;$$
(8)$$\;P_{x}=\sqrt{\frac{1}{n-1}\overset{n}{\underset{i=1}{\sum }}{\left(x_{i}-\bar{x}\right)}^{2}}\;$$
(9)$$\;P_{y}=\sqrt{\frac{1}{n-1}\overset{n}{\underset{i=1}{\sum }}{\left(y_{i}-\bar{y}\right)}^{2}}\;$$
(10)$$\bar{x}=\frac{1}{n}\;\overset{n}{\underset{i=1}{\sum }}x_{i}\;$$
(11)$$\bar{y}=\frac{1}{n}\;\overset{n}{\underset{i=1}{\sum }}y_{i}\;$$
where n is the number of gas sensors, and $x_{i}$ and $y_{i}$ are the *i*th elements of the vector $V_{train}$ and $V_{test}$, respectively. The value of PCC varies from 0 to 1. For k n-dimensional trained vectors, we will obtain k PCCs and k corresponding n-dimensional test vectors. We then choose the biggest PCC and take their corresponding ID as the recognition result.

## 4. Discussion

We have trained the proposed SNN including 24 excitatory neurons with 45 samples of the gas training dataset, including nine types of gases, and each gas has five different concentrations. Five gas samples were randomly selected for learning. The rest of the 180 samples were used for testing. For each excitatory neuron, the gas data are scaled down and selected using data pre-processing as described in Section 3.1, and then are transformed into continuous spikes using the spike precedence coding approach in Section 3.3. To verify the data pre-processing method, we trained our SNN with the raw data for comparison. The comparison results are shown in Figure 4, from which we find that with different numbers of training samples of each gas, an improvement of more than 19.29% in the overall accuracy rate is obtained when using the pre-processed data.

To obtain an optimal network, we studied the impact of network sizes on gas test accuracy based on the pre-processing method. First, we try to find the test accuracy with different numbers of inhibitory neurons (INs) per excitatory neuron through experiments. From Figure 5a, it is observed that as the number of INs increases, so does the test accuracy. The highest accuracy of 96.25% is obtained with five INs per excitatory neuron (ENs) (i.e., the total neuron number is 24@120 neurons). We then needed to find the accuracies with different numbers of synapses, including excitatory and inhibitory synapses of the network with five INs per EN. We changed the connection probability between the excitatory and the inhibitory layers, and the simulation results are illustrated in Figure 5b,c, from which we found that when the connection probabilities of ENs-INs and INs-ENs are 0.8 and 0.25 respectively, the highest test accuracy of 98.75% is achieved.

Although most artificial neural networks suffer from catastrophic forgetting [32], which means they have worse performance after they learn a new type of gas, our network exhibits an online incremental learning ability, and only a small loss of accuracy is seen for the trained gas after a new type of gas is learned. To verify the performance, we built a SNN with 24 ENs, five INs for each EN, about 96 excitatory synapses with the 0.8 connection probability of ENs-INs, and about 690 inhibitory synapses with the 0.25 connection probability of INs-ENs.

We randomly selected a sample for each gas as the training set, and trained the proposed network with one-shot learning gas one by one. After a new gas was learned, we tested the accuracy for the new trained gas, and the results are shown as illustrated in Figure 6. For contrast, we trained the proposed network with one-shot learning on the same training set, and tested the accuracy of each trained gas. The results are shown in Figure 7. By comparing Figure 6 and Figure 7, we find that before training the fifth gas, there is no accuracy loss for all the trained gases: methane, carbon monoxide, hydrogen sulfide, and acetone. For the sixth to ninth trained gas, the accuracy losses are 1.9%, 0.7%, 4.9%, and 1.1%, respectively. Although the recognition accuracies of methanol and ethanol are as low as 33.33% and 34.61%, respectively, there is only a 0.95% overall test accuracy decline after training the gases one by one, compared with training all nine gases with one-shot learning. The optimistic experimental results show that even with one-shot learning, the proposed network can effectively overcome the challenge of catastrophic forgetting which leads to a significant drop in accuracy after learning a new gas.

To further avoid the decrease in accuracy after learning a new gas, we increased the training samples for each gas instead of using one-shot learning. We trained the network with seven other few-shot learning schemes: two-shot, three-shot, and up to eight-shot. First, we needed to find out the relationship between the accuracy loss of all the trained gases after training a new one, and the number of training samples for each gas. As illustrated in Table 2, we found that there is no loss for few-shot learning schemes. This is meaningful because no accuracy loss when learning a new gas proves that our network is able to overcome the sensor drift and aging issues [27]. Once the response of the gas sensors to a specific gas varies greatly, the network will be retrained to keep the accuracy with an incremental learning ability to avoid catastrophic forgetting.

In addition, to improve the overall accuracy of the test set, we retrained the network with different learning schemes. The simulation results are shown in Table 3, from which we can see that with the increase in training samples, the accuracies for all gases rapidly rise and then increase slowly, eventually reaching and maintaining a constant value after training five samples for each gas. In other words, the five-shot learning scheme is the best with the highest accuracy of 98.75%. Moreover, the scheme had a 100% accuracy for methane, carbon monoxide, hydrogen sulfide, ammonia, formaldehyde, and propane. It can be concluded that the proposed network does not underfit or overfit during the testing process.

For further analysis, we collected the correct and incorrect samples and drew a normalized confusion matrix for the accuracies of the nine types of gases in Figure 8; where the horizontal coordinate represents the true class of the gas and the vertical coordinate stands for the predicted type of the gas. As shown in Figure 6, the correct recognition is achieved for most test samples, only a few samples are recognized as another gas: one methanol sample is identified as carbon monoxide and two ethanol samples are recognized as methanol.

Furthermore, we found that our network could identify different types of gases with the same or similar concentrations, but when the test gases have different concentrations, misidentification may happen. For example, the methanol sample at 10 ppm is recognized as carbon monoxide at 300 ppm. Consequently, when some gas sensors are affected by external surroundings, such as the temperature and humidity, small changes in the gas data may lead to changes in concentration, which makes it possible of the network to identify the gas as another gas. The solution to these issues is to enlarge the sensor array. As more sensors means more dimensional characteristics of the gas data, when some sensors are disturbed or even down, other undisturbed dimensions of data characteristics will have a positive effect on the recognition.

We compared the proposed SNN using five-shot learning with six other gas recognition methods: SVM, KNN, DT, PCA + KNN, PCA + SVM, and ANN. We use the radial basis function (RBF) as the kernel function of the SVM model, set K = 9 for the KNN model, and designed the single binary tree with 24 depths according to the differences in response values for each gas sensor, for each gas. The PCA approach is adopted for feature extraction, which reduces the dimensionality of the data by half and then combines it with KNN and SVM classifiers, respectively. Finally, we developed a three-layer artificial neural network using backpropagation with 24 input neurons, 12 hidden neurons, and nine output neurons.

We randomly selected five samples of each gas as a group. Then we adopted the five-fold cross-validation method, that is, one of five groups was taken as the training set in turn, and the others were used as a test set. The accuracy simulation results are shown in Figure 9, from which we can see that compared with SVM, KNN, and DT, better recognition results are obtained with PCA + SVM, PCA + KNN, and ANN. However, PCA + SVM and PCA + KNN performed poorly in the recognition of methane with 76% and 78% accuracy rates, respectively, methanol with 67% and 62%, respectively, formaldehyde with 66% and 73%, respectively, and ethanol with 73% and 73%, respectively, and the ANN model recognized ethanol with an accuracy of 83%. However, as shown in Figure 8, the proposed SNN had a 92% accuracy rate in the recognition of ethanol, which is 9% higher than that of ANN. Furthermore, our work had a 100% accuracy rate in the gas recognition of methane, carbon monoxide, hydrogen sulfide, acetone, ammonia, formaldehyde, and propane. Other methods such as PCA+SVM could only identify less than three gases with a 100% accuracy.

The overall accuracy for comparison between the proposed network and other methods is shown in Figure 10, from which we found that the proposed network has the highest accuracy of 98.75%, better than the SVM model with a 56.33% accuracy, the KNN model with 62.22%, the DT with 51.00%, the PCA + SVM with 83.11%, the PCA + KNN with 84.55%, and the ANN with 93.66%. Compared with the common (SVM, KNN, DT, PCA, and ANN) gas recognition algorithms, our work achieved an accuracy improvement of up to 5.09%. The proposed network had a better classification performance on cross-concentration gas recognition with the gas dataset for nine types of gases. Furthermore, the proposed network with the five-shot learning scheme had no accuracy loss when learning a new gas, thereby presenting incremental online learning ability.

To better present and validate the performance of the model, we built a deep learning ANN model using eight layers and a CNN model using 10 layers, by experimentally tuning the network size and parameters. The experimental results show that the deep learning models cannot obtain satisfactory accuracy results (85.7% with the 8-layer ANN and 89.3% with the 10-layer CNN). The reason many be that in the case of limited gas sensors (24 gas sensor in our work), the low-dimensional input data make the synaptic weights undergo gradient vanishing or gradient exploding during the weight update iterations, making the deep network over fit or under fit, resulting in a low recognition accuracy.

To further verify the performance of the proposed SNN, we tested our algorithm on the publicly available UCSD gas sensor drift dataset [32,33], which contains 13,910 gas samples from an array of 16 chemo sensors (eight features per sensor), exposed to six types of gas (ammonia, acetaldehyde, acetone, ethylene, ethanol, and toluene), and distributed across 10 batches that were sampled over three years to emphasize the challenge of sensor drift over time. For every batch of the UCSD gas sensor drift dataset, we randomly split the gas samples into a training set and a test set in a 4:1 ratio. The test accuracy results are shown in Table 4, from which we find the proposed network has the best average accuracy of 99.26% on the UCSD gas sensor drift dataset, which has a 4.33% average accuracy improvement to that of [32]. The optimistic results reveal that the proposed network has a competitive gas recognition performance.

## 5. Conclusions

In this work, a bio-inspired spiking neural network inspired by biological olfaction is proposed to recognize various types of flammable and toxic gases. We adopt a variety of min-max normalizations for data preprocessing method for feature extraction and interference resistance. Based on STDP, we innovatively designed an SNN with the inhibitory mechanism to further highlight the spike-phase relationship of gas data in different channels to achieve the winner-takes-all effect. The comparison with other algorithms showed that our network achieved the highest accuracy of 98.75% in a five-fold cross-validation for the identification of nine gases, each with five different concentrations. In addition, our network supports incremental learning, which is helpful to address sensor drifting and aging. The experimental results show that there is no accuracy loss for all trained gases after training a new one. To summarize, the proposed network with few-shot class-incremental learning is a feasible approach for fast detection and noise resistance, which is meaningful in real-life fire scenarios. In the future, we will apply the proposed network to more applications as well as E-noses to access its versatility further. We envisage that this can provide a universal gas recognition method. Our objective is also to improve the structure of the neural network by, for example, adding additional classifiers, which will enable the network to perform better recognition.

## Figures and Tables

**Figure 1 sensors-23-02433-f001:**
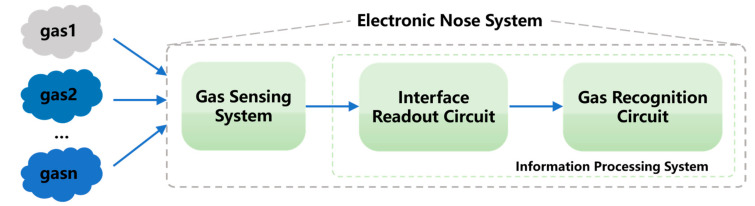
E-nose system framework.

**Figure 2 sensors-23-02433-f002:**
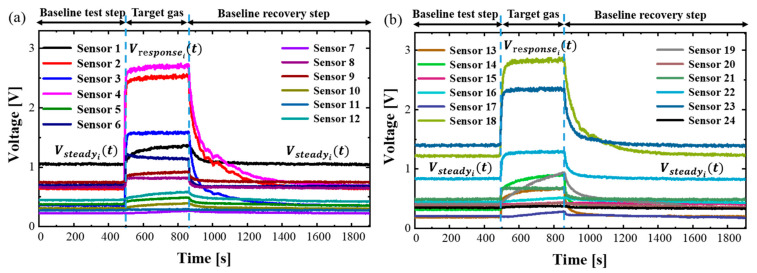
The response curve of 24 gas sensors with 2000 ppm CH4. (**a**) Response curves of the first twelve chemical gas sensors; (**b**) Response curves of the last twelve chemical gas sensors.

**Figure 3 sensors-23-02433-f003:**
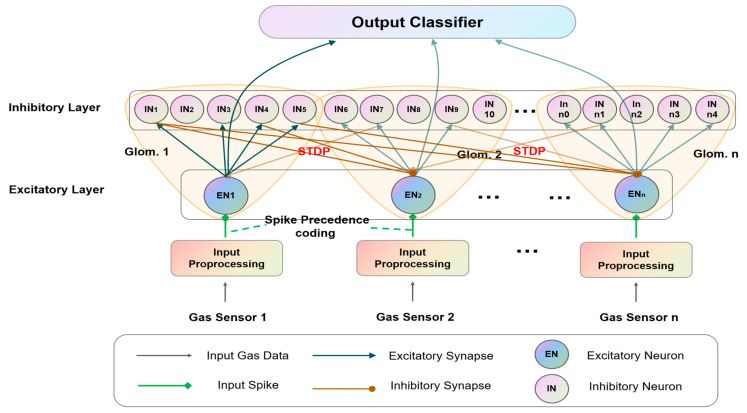
Schematic model of the proposed spiking neural network.

**Figure 4 sensors-23-02433-f004:**
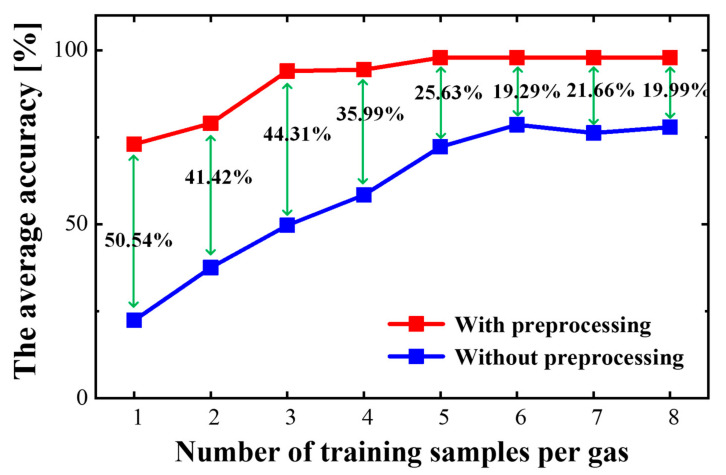
The average accuracy for all 9 types of gas with different numbers of training samples for each gas.

**Figure 5 sensors-23-02433-f005:**
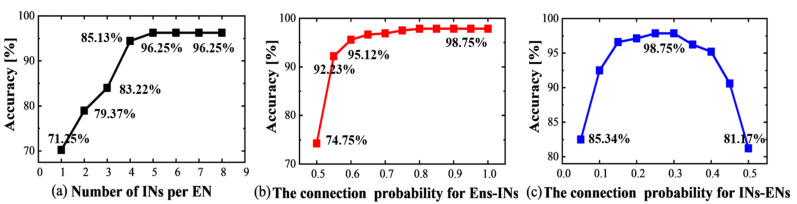
(**a**) The overall accuracy for various numbers of INs (Inhibitory Neurons) for each EN (Excitatory Neuron); (**b**) The overall accuracy for different connection probabilities for ENs-INs; (**c**) The overall accuracy for the different connection probabilities for INs-ENs.

**Figure 6 sensors-23-02433-f006:**
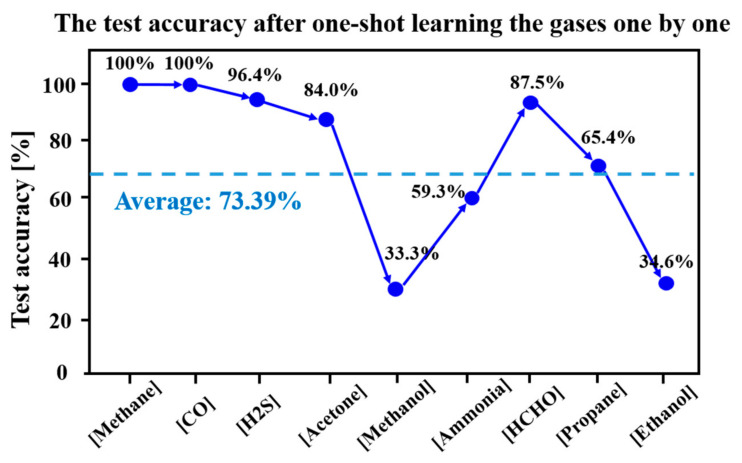
The test accuracy after one-shot learning the gases one by one.

**Figure 7 sensors-23-02433-f007:**
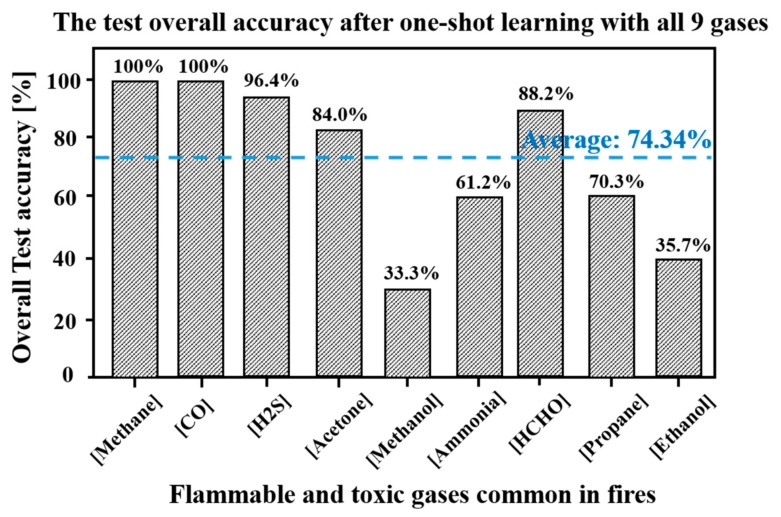
The overall test accuracy after one-shot learning gases with all 9 gases.

**Figure 8 sensors-23-02433-f008:**
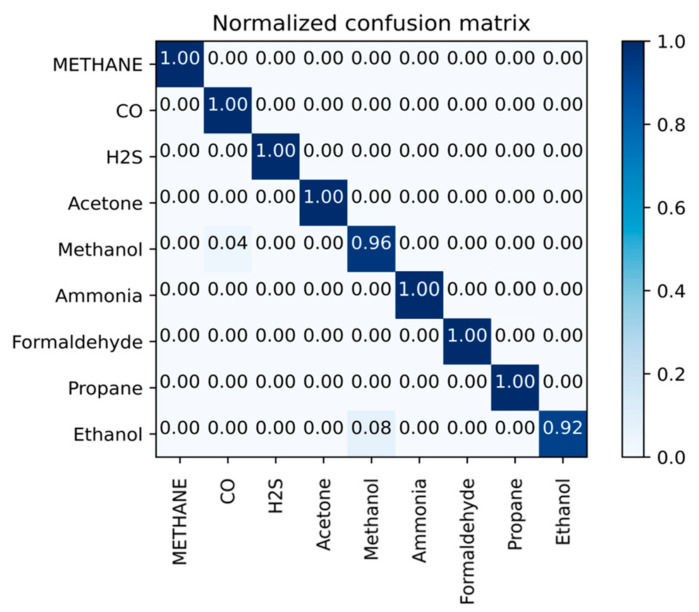
The confusion matrix for the accuracy of nine types of gases with the five-shot learning scheme.

**Figure 9 sensors-23-02433-f009:**
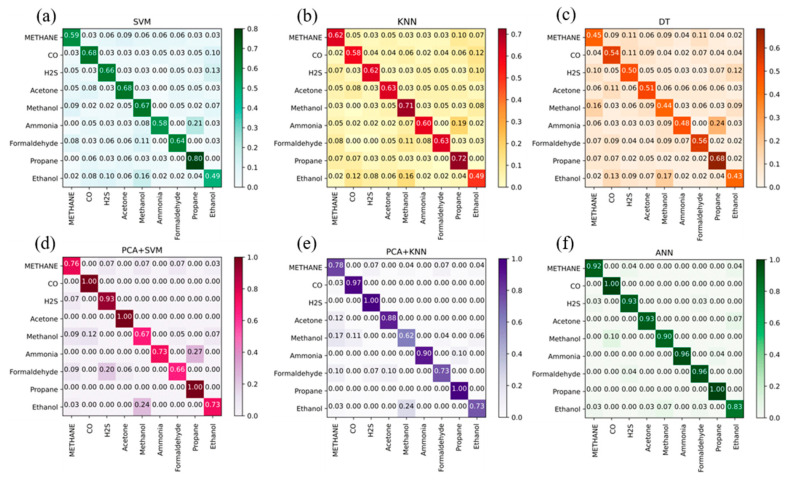
The confusion matrix for the accuracies of (**a**) SVM; (**b**) KNN; (**c**) DT; (**d**) PCA+SVM; (**e**) PCA + KNN; (**f**) ANN.

**Figure 10 sensors-23-02433-f010:**
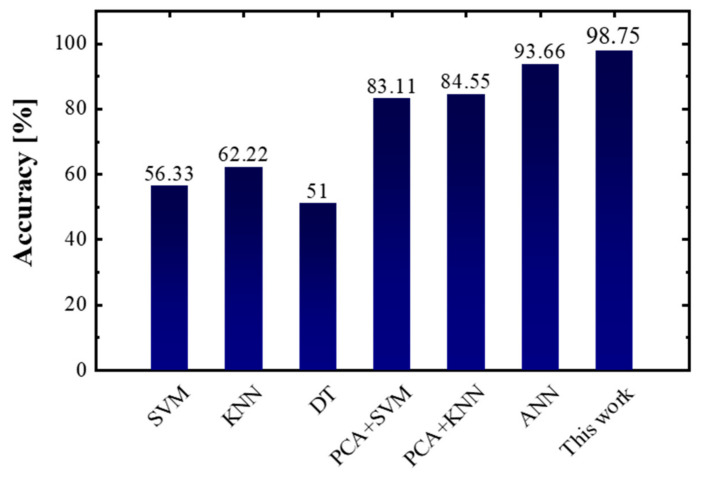
The overall accuracy of all the gas recognition methods for comparison.

**Table 1 sensors-23-02433-t001:** Analyses and their concentration used in this work.

Analyses	Concentration Levels (ppm)	Flammability	Toxicity
Formaldehyde	0.10, 0.15, 0.30, 0.50, 0.70	Low	Moderate
Ethanol	50, 100, 200, 500, 1000	Low	Low
Propane	200, 500, 800, 1000, 1500	Low	Low
Methanol	10, 20, 50, 80, 100	Moderate	Moderate
Methane	1000, 2000, 3000, 5000, 8000	Low	High
Carbon monoxide	100, 150, 200, 250, 300	Moderate	Moderate
Acetone	10, 20, 50, 80, 100	Low	Low
Hydrogen sulfide	0.20, 0.50, 0.80, 1.00, 1.50	Low	High
Ammonia	5.0, 10.0, 20.0, 50.0, 80.0	Moderate	Moderate

**Table 2 sensors-23-02433-t002:** The test accuracy of the proposed SNN with two different learning methods.

Overall Test Accuracy	One-Shot	Two-Shot	Three-Shot	Four-Shot	Five-Shot	Six-Shot	Seven-Shot	Eight-Shot
Learning all 9 gases (%)	74.34 ± 1.6	78.97 ± 1.3	93.99 ± 0.5	94.42 ± 0.3	98.75 ± 0.1	98.75 ± 0.1	98.75 ± 0.1	98.75 ± 0.1
Learning gases one by one (%)	73.39 ± 2.1	78.97 ± 1.2	93.99 ± 0.9	94.42 ± 0.5	98.75 ± 0.1	98.75 ± 0.1	98.75 ± 0.1	98.75 ± 0.1
Accuracy loss (%)	0.95	0.00	0.00	0.00	0.00	0.00	0.00	0.00

**Table 3 sensors-23-02433-t003:** The test accuracy for different gases based on the proposed SNN with different few-shot learning methods.

Accuracy (%)	One-Shot	Two-Shot	Three-Shot	Four-Shot	Five-Shot	Six-Shot	Seven-Shot	Eight-Shot
Methane	100 ± 0.0	100 ± 0.0	100 ± 0.0	100 ± 0.0	100 ± 0.0	100 ± 0.0	100 ± 0.0	100 ± 0.0
Carbon monoxide	100 ± 0.0	100 ± 0.0	100 ± 0.0	100 ± 0.0	100 ± 0.0	100 ± 0.0	100 ± 0.0	100 ± 0.0
Hydrogen sulfide	96.42 ± 1.3	100 ± 0.0	100 ± 0.0	100 ± 0.0	100 ± 0.0	100 ± 0.0	100 ± 0.0	100 ± 0.0
Acetone	84.00 ± 3.2	76.00 ± 1.9	92.00 ± 1.6	100 ± 0.0	100 ± 0.0	100 ± 0.0	100 ± 0.0	100 ± 0.0
Methanol	33.33 ± 5.1	74.07 ± 1.5	92.59 ± 1.3	81.48 ± 1.2	96.29 ± 0.1	96.29 ± 0.1	96.29 ± 0.1	96.29 ± 0.1
Ammonia	59.25 ± 2.1	66.67 ± 2.1	92.59 ± 1.3	100 ± 0.0	100 ± 0.0	100 ± 0.0	100 ± 0.0	100 ± 0.0
Formaldehyde	87.50 ± 1.9	62.50 ± 1.5	100 ± 0.0	100 ± 0.0	100 ± 0.0	100 ± 0.0	100 ± 0.0	100 ± 0.0
Propane	65.38 ± 2.2	84.62 ± 1.6	100 ± 0.0	100 ± 0.0	100 ± 0.0	100 ± 0.0	100 ± 0.0	100 ± 0.0
Ethanol	34.61 ± 3.1	46.15 ± 2.8	69.23 ± 3.9	69.23 ± 3.3	92.31 ± 0.8	92.31 ± 0.8	92.31 ± 0.8	92.31 ± 0.8
Average	73.39 ± 2.1	78.97 ± 1.2	93.99 ± 0.9	94.42 ± 0.5	98.75 ± 0.1	98.75 ± 0.1	98.75 ± 0.1	98.75 ± 0.1

**Table 4 sensors-23-02433-t004:** The test accuracy comparisons of the publicly available UCSD gas sensor drift dataset.

Accuracy (%)	Batch1	Batch2	Batch3	Batch4	Batch5	Batch6	Batch7	Batch8	Batch9	Batch10	Average
SNN’2019 [32]	99.39	92.44	94.95	97.73	98.22	94.55	89.74	92.30	99.48	90.46	94.93
MLP’2021 [34]	95.63	95.42	94.53	99.56	99.20	90.27	89.96	96.50	98.11	80.81	94.00
This work	98.88 ± 0.3	99.80 ± 0.0	99.89 ± 0.0	100 ± 0.0	100 ± 0.0	98.91 ± 0.1	99.38 ± 0.1	98.31 ± 0.0	98.94 ± 0.0	98.47 ± 0.0	99.26 ± 0.0

## Data Availability

Not applicable.

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
