# Peer review of "A Bio-Inspired Spiking Neural Network with Few-Shot Class-Incremental Learning for Gas Recognition"

_sensors, 2023, doi:10.3390/s23052433_

Round 1

Reviewer 1 Report

there is a space between the number and the unit.

lines 125-128, the equations should be made in brief.

effect of the timing constant ? on acc?

pls provide the information of train data in detail

how many train data, test data. Are those selected in random. Recommendation, the processes of selecting data, train and test shall be made in many times. then, Standard deviation of acc should be provided in the Table 1, 3 and 4

Author Response

Please see the attachment which includes the revised paper and the corresponding response letter.

Response Letter 

Reviewer: 1

Comments to the Author

  1. There is a space between the number and the unit.

----Response:

Thank you for the valuable suggestion. We have added the space between the number and the unit.

  1. lines 125-128, the equations should be made in brief.

----Response:

Thank you for your comments. We have simplified the equations and descriptions in lines 125-128 to make the paper more concise.

Related changes to the manuscript have been highlighted in yellow color in the latest version of the paper.

  1. effect of the timing constant ? on acc?

----Response:

Thank you for the valuable comment.

We adopt the integrate-and-fire (IF) model to simulate the basic functions of biological neurons. According to the reference [1], ? is taken as the timing constant of the neuron membrane, modeling the voltage leakage.

If the ? is too large, the voltage leakage of the neurons in our spiking neural network (SNN) also increases, thereby the membrane voltage  is hard to reach the membrane threshold  , therefore the neuron is also hard to spike. Consequently, the neuron in the second layer cannot receive enough spikes from the first layer to spike. As a result, the network cannot work due to the too-low spike activities.

On the other hand, if the ? is too small, the voltage leakage of the neurons will decrease, which makes all neurons easy to spike. Hence,  the main features of input spikes are hard to be extracted correctly by the SNN. That is, the too-small ? will directly reduce the recognition accuracy.

 [1] Paugam-Moisy, H., Bohte, S. (2012). Computing with Spiking Neuron Networks. In: Rozenberg, G., Bäck, T., Kok, J.N. (eds) Handbook of Natural Computing. Springer, Berlin, Heidelberg. https://doi.org/10.1007/978-3-540-92910-9_10

  1. pls provide the information of train data in detail

----Response:

Thank you for the valuable suggestion.

In Section 2, based on the E-nose gas sampling system from Suzhou Huiwen Nanotechnology Co., Ltd., we collected 225 gas samples covering the following 9 types of flammable and explosive gases: formaldehyde, ethanol, propane, methanol, methane, carbon monoxide, acetone, hydrogen sulfide, and ammonia. As shown in Table 1, each gas was collected across 5 concentration levels and 5 gas samples were collected independently at each concentration level, resulting in 25 samples per gas and a total of 225 gas samples for 9 gases.

We randomly chose 5 samples out of the 25 samples for each gas as training samples, thereby a total of 45 samples for 9 types of gases. These 45 samples are used as the training set. The train data information of each gas sample is shown in Fig.1, where the horizontal coordinate is the response voltage value (v), and the vertical coordinate is the time (s). The gas data processing methods are detailed in Section 2.

Table 1. Analyses and their concentration used in this work.

Analyses

Concentration Levels (ppm)

 Flammability

Toxicity

Formaldehyde

0.10, 0.15, 0.30, 0.50, 0.70

Low

Moderate

Ethanol

50, 100, 200, 500, 1000

Low

Low

Propane

200, 500, 800, 1000, 1500

Low

Low

Methanol

10, 20, 50, 80, 100

Moderate

Moderate

Methane

1000, 2000, 3000, 5000, 8000

Low

High

Carbon monoxide

100, 150, 200, 250, 300

Moderate

Moderate

Acetone

10, 20, 50, 80, 100

Low

Low

Hydrogen sulfide

0.20, 0.50, 0.80, 1.00, 1.50

Low

High

Ammonia

5.0, 10.0, 20.0, 50.0, 80.0

Moderate

Moderate

  1. how many train data, test data. Are those selected in random. Recommendation, the processes of selecting data, train and test shall be made in many times. then, Standard deviation of acc should be provided in the Table 1, 3 and 4

----Response:

Thank you for your comments and recommendation. According to the answer to question 4, there are 45 training data and 180 test data, and these data are selected in random. According to your suggestion, for each gas sample data, we randomly select 10 values for each of the baseline test step and the target gas test step,and process them according to the formula (1) and repeat 10 times to get the average value as the network input.

 (1)  Since Table 1 is not involved in the accuracy, we have added a standard deviation of accuracy in Tables 2-4.

Table 2. The test accuracy of the proposed SNN with two different learning methods

Test Overall Accuracy

One-shot

Two-shot

Three-shot

Four-shot

Five-shot

Six-shot

Seven-shot

Eight-shot

Learning all 9 gas (%)

74.34±1.6

78.97±1.3

93.99±0.5

94.42±0.3

98.75±0.1

98.75±0.1

98.75±0.1

98.75±0.1

Learning gas one by one (%)

73.39±2.1

78.97±1.2

93.99±0.9

94.42±0.5

98.75±0.1

98.75±0.1

98.75±0.1

98.75±0.1

Accuracy loss (%)

0.95

0.00

0.00

0.00

0.00

0.00%

0.00

0.00

Table 3. The test accuracy for different gases based on the proposed SNN with different few-shot learning methods.

Accuracy (%)

One-shot

Two-shot

Three-shot

Four-shot

Five-shot

Six-shot

Seven-shot

Eight-shot

Methane

100±0.0

100±0.0

100±0.0

100±0.0

100±0.0

100±0.0

100±0.0

100±0.0

Carbon monoxide

100±0.0

100±0.0

100±0.0

100±0.0

100±0.0

100±0.0

100±0.0

100±0.0

Hydrogen sulfide

96.42±1.3

100±0.0

100±0.0

100±0.0

100±0.0

100±0.0

100±0.0

100±0.0

Acetone

84.00±3.2

76.00±1.9

92.00±1.6

100±0.0

100±0.0

100±0.0

100±0.0

100±0.0

Methanol

33.33±5.1

74.07±1.5

92.59±1.3

81.48±1.2

96.29±0.1

96.29±0.1

96.29±0.1

96.29±0.1

Ammonia

59.25±2.1

66.67±2.1

92.59±1.3

100±0.0

100±0.0

100±0.0

100±0.0

100±0.0

Formaldehyde

87.50±1.9

62.50±1.5

100±0.0

100±0.0

100±0.0

100±0.0

100±0.0

100±0.0

Propane

65.38±2.2

84.62±1.6

100±0.0

100±0.0

100±0.0

100±0.0

100±0.0

100±0.0

Ethanol

34.61±3.1

46.15±2.8

69.23±3.9

69.23±3.3

92.31±0.8

92.31±0.8

92.31±0.8

92.31±0.8

Average

73.39±2.1

78.97±1.2

93.99±0.9

94.42%±0.5

98.75±0.1

98.75±0.1

98.75±0.1

98.75±0.1

Table 4. The test accuracy comparisons of the publicly available UCSD gas sensor drift dataset

Accuracy (%)

Batch1

Batch2

Batch3

Batch4

Batch5

Batch6

Batch7

Batch8

Batch9

Batch10

Average

SNN’2019 [32]

99.39

92.44

94.95

97.73

98.22

94.55

89.74

92.30

99.48

90.46

94.93

MLP’2021 [36]

95.63

95.42

94.53

99.56

99.20

90.27

89.96

96.50

98.11

80.81

94.00

This work

98.88±0.3

99.80±0.0

99.89±0.0

100±0.0

100±0.0

98.91±0.1

99.38±0.1

98.31±0.0

98.94±0.0

98.47±0.0

99.26±0.0

Reviewer 2 Report

In the given manuscript the authors have developed an AI-based 

a bio-inspired spiking neural network (SNN) for gas recognition. The authors have claimed that the proposed model can recognize the gas even in the variation of the sensor's selectivity and sensitivity profile. 

Various researchers in literature have proposed their model for gas recognition using this own developed DNN model. The author claims to achieve 98.75% recognition accuracy and presented the confusion matrix in the results section. The two layers including the excitatory layer connected with one gas sensor brought robustness to the recognition performance. The work proposed by the authors is quite interesting. However, I found a few discrepancies that require to be tackled before publication.

1) The author has presented his method for gas recognition which has achieved promising results; however, the author has not compared the results with similar works in the literature works for the validation of his work. The comparison shown in figure 10 does not lie in the domain of the neural model. The SVM, KNN, and DT are the classification models not need to be used for the extraction of features; whereas the SNN model used by the author is a neural network capable of extracting and classifying feature data. To present and validate the performance of the model the author needs to present the Deep learning models in this comparison section. 

2) The author has not provided any information related to the availability of the dataset for future research on the same problem. The method list of gas data can be generated; however, the sensors profile, aging factor, and environmental challenges are hard for the reader/researcher to add to the data; therefore we recommend providing data for further research. Furthermore; if the authors upload their model on GitHub or any other public database this would help the researchers in general.

Author Response

Please see the attachment which includes the revised paper and the response letter.

Response Letter

Reviewer: 2

Comments to the Author

In the given manuscript the authors have developed an AI-based a bio-inspired spiking neural network (SNN) for gas recognition. The authors have claimed that the proposed model can recognize the gas even in the variation of the sensor's selectivity and sensitivity profile.

Various researchers in literature have proposed their model for gas recognition using this own developed DNN model. The author claims to achieve 98.75% recognition accuracy and presented the confusion matrix in the results section. The two layers including the excitatory layer connected with one gas sensor brought robustness to the recognition performance. The work proposed by the authors is quite interesting. However, I found a few discrepancies that require to be tackled before publication.

Thank you for your review of our paper. We have answered each of your points below, and revised the paper accordingly. The changes are highlighted in yellow color.

  1. The author has presented his method for gas recognition which has achieved promising results; however, the author has not compared the results with similar works in the literature works for the validation of his work. The comparison shown in figure 10 does not lie in the domain of the neural model. The SVM, KNN, and DT are the classification models not need to be used for the extraction of features; whereas the SNN model used by the author is a neural network capable of extracting and classifying feature data. To present and validate the performance of the model the author needs to present the Deep learning models in this comparison section.

----Response:

Thank you for your comments and suggestions. In our work, we have chosen SVM, KNN, PCA+SVM, PCA+KNN and ANN as our comparison approaches because these methods are classical in the field of gas recognition and have been implemented successfully in many gas recognition applications. Many reviews [1-3] and research [4-7] on E-noses can offer evidence for our comment. Therefore, we compare these six methods with our work based on our dataset. 

Although the comparison shown in Fig.10 does not line in the domain of other SNN algorithms, we compare our work with other gas recognition methods including ANN. According to the study [8], SNNs still lag behind ANNs in terms of accuracy. Therefore, we believe that comparing the accuracy of our work with that of ANN directly can prove the accuracy advantage of our work. In addition, we have tried to adopt different neuronal models including the integrate-and-fire (IF) model, the leaky integrate-and-fire (LIF) model, and the spiking response model (SRM) to build our SNN. But we find that the accuracy of gas recognition using the IF model is significantly higher than the accuracy of the other two. Therefore, we choose the IF model as our neural model.

IF model

LIF model

SRM model

Average accuracy

98.75%

82.31%

31.56%

Yes, the SVM, KNN, and DT are the classification models not need to be used for the extraction of features, but our SNN model includes the preprocessing module used to extract features. Therefore, to compare the accuracy more fairly, we introduce combination models: PCA+SVM and PCA+KNN, in which PCA is used for data feature extraction [1]. As expected, the combination models have achieved higher accuracy in gas identification applications [9-10]. Thus, we believe it is meaningful to compare the accuracy of our SNN model with that of the combination models of PCA+SVM, and PCA+KNN.

 According to your suggestion, to better present and validate the performance of the model, we build a deep learning ANN model using 8 layers (24-18-24-18-32-24-12-9) and a CNN model using 10 layers (Input layer (24*4*1) -Convolutional layer1(24*4*32) - Convolutional layer2 (24*4*32) - Pooling layer - Convolutional layer3 (12*2*64) - Convolutional layer4 (12*2*64) - Pooling layer (1*1*64)- 3 layer fully-connected network (64-32-9)) by experimentally tuning the network size and parameters. The experimental results show that using the deep learning models cannot obtain satisfying accuracy results (85.7% with 8-layer ANN and 89.3% with 10-layer CNN). The reason many be that in the case of limited gas sensors (24 gas sensor in our work), the low dimensional input data makes the synaptic weight undergo gradient vanishing or gradient exploding during the weight update iterations, making the deep network over fit or under fit, and results in low recognition accuracy.

SVM

KNN

DT

PCA
+SVM

PCA
+KNN

ANN
(3-layer)

Deep ANN
(8-layer)

Deep CNN
(10-layer)

Our work

Acc.(%)

56.33

62.22

51

83.11

84.55

93.66

85.72

89.28

98.75

The revisions concerned are highlighted as yellow color in the latest version of the paper in section 4.

[1] H. Chen, D. Huo and J. Zhang, "Gas Recognition in E-Nose System: A Review," in IEEE Transactions on Biomedical Circuits and Systems, vol. 16, no. 2, pp. 169-184, April 2022, doi: 10.1109/TBCAS.2022.3166530.

[2] J. Tan and J. Xu, "Applications of electronic nose (e-nose) and electronic tongue (e-tongue) in food quality-related properties determination: A review," Artificial Intelligence in Agriculture, vol. 4, pp. 104-115,2020

[3] Cheng, L., Meng, Q. H., Lilienthal, A. J., & Qi, P. F. (2021). Development of compact electronic noses: A review. Measurement Science and Technology, 32(6), 062002.

[4] V. Schroeder et al., "Chemiresistive sensor array and machine learning classification of food," ACS sensors, vol. 4, no. 8, pp. 2101-2108, 2019.

[5] S. Qiu and J. Wang, "The prediction of food additives in the fruit juice based on electronic nose with chemometrics," Food chemistry, vol. 230, pp. 208-214, 2017.
[6] W. Harsono, R. Sarno, and S. I. Sabilla, "Recognition of Original Arabica Civet Coffee based on Odor using Electronic Nose and Machine Learning," in 2020 International Seminar on Application for Technology of Information and Communication (iSemantic), 2020: IEEE, pp. 333-339.

[7] E. Mirzaee-Ghaleh, A. Taheri-Garavand, F. Ayari, and J. Lozano, "Identification of fresh-chilled and frozen-thawed chicken meat and estimation of their shelf life using an E-nose machine coupled fuzzy KNN," Food Analytical Methods, vol. 13, no. 3, pp. 678-689, 2020.

[8] A. Tavanaei, M. Ghodrati, S. R. Kheradpisheh, T. Masquelier, and A. Maida, "Deep learning in spiking neural networks," Neural Netw, vol. 111, pp. 47-63, 2019.

[9] L.-Y. Chen, C.-C. Wu, T.-I. Chou, S.-W. Chiu, and K.-T. Tang, "Development of a Dual MOS electronic nose/camera system for improving fruit ripeness classification," Sensors, vol. 18, no. 10, p. 3256, 2018.

[10] W. Jia, G. Liang, H. Tian, J. Sun, and C. Wan, "Electronic nose-based technique for rapid detection and recognition of moldy apples," Sensors, vol. 19, no. 7, p. 1526, 2019.

  1. The author has not provided any information related to the availability of the dataset for future research on the same problem. The method list of gas data can be generated; however, the sensors profile, aging factor, and environmental challenges are hard for the reader/researcher to add to the data; therefore we recommend providing data for further research. Furthermore; if the authors upload their model on GitHub or any other public database this would help the researchers in general.

----Response:

Thank you for your comment and suggestion.

We will take your valuable suggestion fully into account. Compared with the publicly available gas datasets [11-13], our gas sensor dataset is not complete enough so far with only 225 samples of 9 flammable and explosive gases. But, we will continue to collect more experimental samples of other flammable and explosive gases and mixed gases. When we collect enough gas samples, we will collate all the gas samples into a publicly available dataset and upload it together with our model on GitHub later.

[11] A. Vergara, S. Vembu, T. Ayhan, M. A. Ryan, M. L. Homer and R. Huerta, "Chemical gas sensor drift compensation using classifier ensembles", Sens. Actuators B Chem., vol. 166, pp. 320-329, May 2012.

[12] I. Rodriguez-Lujan, J. Fonollosa, A. Vergara, M. Homer and R. Huerta, "On the calibration of sensor arrays for pattern recognition using the minimal number of experiments", Chemometrics Intell. Lab., vol. 130, pp. 123-134, Jan. 2014.

[13] Ramon Huerta, Thiago Mosqueiro, Jordi Fonollosa, Nikolai Rulkov, Irene Rodriguez-Lujan. Online Decorrelation of Humidity and Temperature in Chemical Sensors for Continuous Monitoring. Chemometrics and Intelligent Laboratory Systems 2016.
